# BALANCING COST AND BENEFIT WITH TIED-MULTI TRANSFORMERS

## ABSTRACT

This paper proposes a novel procedure for training multiple Transformers with tied parameters which compresses multiple models into one enabling the dynamic choice of the number of encoder and decoder layers during decoding. In sequence-to-sequence modeling, typically, the output of the last layer of the $N$-layer encoder is fed to the $M$-layer decoder, and the output of the last decoder layer is used to compute loss. Instead, our method computes a single loss consisting of $N \times M$ losses, where each loss is computed from the output of one of the $M$ decoder layers connected to one of the $N$ encoder layers. A single model trained by our method subsumes multiple models with different number of encoder and decoder layers, and can be used for decoding with fewer than the maximum number of encoder and decoder layers. We then propose a mechanism to choose a priori the number of encoder and decoder layers for faster decoding, and also explore recurrent stacking of layers and knowledge distillation to enable further parameter reduction. In a case study of neural machine translation, we present a cost-benefit analysis of the proposed approaches and empirically show that they greatly reduce decoding costs while preserving translation quality.

## 1 INTRODUCTION

Neural networks for sequence-to-sequence modeling typically consist of an encoder and a decoder coupled via an attention mechanism. Whereas the very first deep models used stacked recurrent neural networks (RNN) (Sutskever et al., 2014; Cho et al., 2014; Bahdanau et al., 2015) in the encoder and decoder, the recent Transformer model (Vaswani et al., 2017) constitutes the current state-of-the-art approach, owing to its better context generation mechanism via multi-head self- and cross-attentions.

Given an encoder-decoder architecture and its hyper-parameters, such as the number of encoder and decoder layers, vocabulary sizes (in the case of text based models) and hidden layers, the parameters of the model, i.e., matrices and biases for non-linear transformations, are optimized by iteratively updating them so that the loss for the training data is minimized. The hyper-parameters can also be tuned, for instance, through maximizing the automatic evaluation score on the development data. However, in general, it is highly unlikely (or impossible) that a single optimized model suffices diverse cost-benefit demands at the same time. For instance, in practical low-latency scenarios, one may accept some performance drop for speed. However, a model used with a subset of optimized parameters might perform badly. A single optimized model cannot also guarantee the best performance for each individual input. Although this is practically important, it has drown only a little attention. An existing solution for this problem is to host multiple models simultaneously for flexible choice. However, this approach is not very practical, because it requires an unreasonably large quantity of resources. Furthermore, there are no well-established methods for selecting appropriate models for each individual input.

As a more effective solution, we consider training a single model that subsumes multiple models which can be used for decoding with different hyper-parameter settings depending on the input or on the latency requirements. In this paper, we focus on the number of layers as an important hyper-parameter that impacts both speed and quality of decoding, and propose a *multi-layer softmaxing* method, which trains multi-layer neural models referring to the output of all layers during training. Conceptually, as illustrated in Figure 1, this method involves tying (sharing) the parameters of mul-

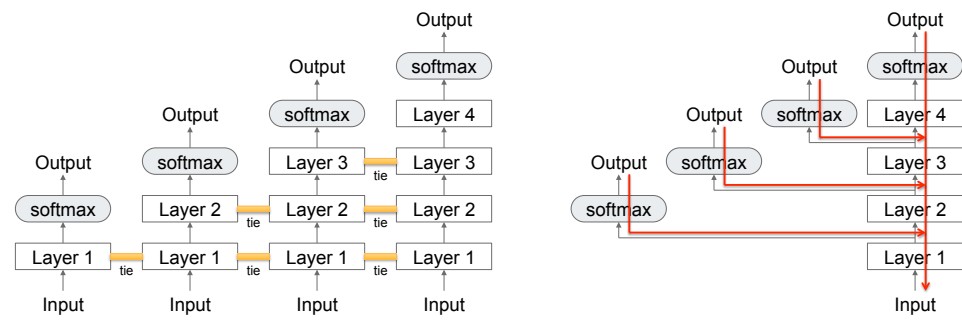

(a) Multiple tied-layer vanilla models.          (b) Collapsing tied layers into one.

Figure 1: The general concept of multi-layer softmaxing for training multi-layer neural models with an example of a 4-layer model. Figure 1a is a depiction of our idea in the form of multiple vanilla models whose layers are tied together. Figure 1b shows the result of collapsing all tied layers into a single layer. The red lines indicate the flow of gradients and hence the lowest layer in the stack receives the largest number of updates.

tiple models with different number of layers and is not specific to particular types of multi-layer neural models. On top of the above method, we consider to exploit the model further. For saving decoding time, we design and evaluate mechanisms to choose appropriate number of layers depending on the input. As for further model compression, we leverage other orthogonal types of parameter tying approaches, such as those reviewed in Section 2.

Despite the generality of our proposed method, in this paper, we particularly focus on encoder-decoder models with $N$ encoder and $M$ decoder layers, and compress $N \times M$ models[1] by updating the model with a total of $N \times M$ losses computed by softmaxing the output of each of the $M$ decoder layers, where it attends to the output of each of the $N$ encoder layers. The number of parameters of the resultant encoder-decoder model is equivalent to that of the most complex subsumed model with $N$ encoder and $M$ decoder layers. Yet, we can now perform faster decoding using a fewer number of layers, given that shallower layers are better trained. To evaluate our proposed method, we take the case study of neural machine translation (NMT) (Cho et al., 2014; Bahdanau et al., 2015), where we focus on the numbers of encoder and decoder layers of the Transformer model (Vaswani et al., 2017), and demonstrate that a single model with $N$ encoder and $M$ decoder layers trained by our method can be used for flexibly decoding with fewer than $N$ and $M$ layers without appreciable quality loss. We evaluate our proposed method on WMT18 English-to-German translation task, and give a cost-benefit analysis for translation quality vs. decoding speed.

The rest of the paper is organized as follows. Section 2 briefly reviews related work for compressing neural models. Section 3 covers our method that ties multiple models by softmaxing all encoder-decoder layer combinations. Section 4 describes our efforts towards designing and evaluating a mechanism for dynamically selecting encoder-decoder layer combinations prior to decoding. Section 5 describes two orthogonal extensions to our model aiming at further model compression and speeding-up of decoding. The paper ends with Section 6 containing conclusion and future work.

## 2    RELATED WORK

There are studies that exploit multiple layers simultaneously. Wang et al. (2018) fused hidden representations of multiple layers in order to improve the translation quality. Belinkov et al. (2017) and Dou et al. (2018) attempted to identify which layer can generate useful representations for different natural language processing tasks. Unlike them, we make all layers of the encoder and decoder usable for decoding with any encoder-decoder layer combination. In practical scenarios, we can save significant amounts of time by choosing shallower encoder and decoder layers for inference.

Our method ties the parameters of multiple models, which is orthogonal to the work that ties parameters between layers (Dabre & Fujita, 2019) and/or between the encoder and decoder within a

---

[1]Rather than casting the encoder-decoder model into a single column model with $(N + M)$ layers.

---

**Algorithm 1** Training a tied-multi model

---

1: $enc_0 = X$
2: **for** $i$ in 1 to $N$ **do**
3:     $enc_i = L_i^{enc}(enc_{i-1})$
4:     **for** $j$ in 1 to $M$ **do**
5:         $dec_j = L_j^{dec}(dec_{j-1}, enc_i)$
6:         $\hat{Y} = softmax(dec_j)$
7:         $loss_{i,j} = cross\_entropy(\hat{Y}, Y)$
8:     **end for**
9: **end for**
10: $overall\_loss = aggregate(loss_{1,1}, \ldots, loss_{N,M})$
11: Back-propagate using $overall\_loss$

---

single model (Xia et al., 2019; Dabre & Fujita, 2019). Parameter tying leads to compact models enabling faster decoding, but they usually suffer from drops in inference quality. In this paper, we counter such drops with knowledge distillation (Hinton et al., 2015; Kim & Rush, 2016; Freitag et al., 2017). The working principle of this approach is to utilize smoothed data or smoothed training signals instead of the actual training data. A model with a large number of parameters and high performance provides smoothed distributions that are then used as labels for training small models instead of one-hot vectors.

As one of the aims in this work is model size reduction, it is related to a growing body of work that addresses the computational requirement reduction. Pruning of pre-trained models (See et al., 2016) makes it possible to discard around 80% of the smallest weights of a model without deterioration in inference quality, given it is re-trained with appropriate hyper-parameters after pruning. Currently, most deep learning implementations use 32-bit floating point representations, but 16-bit floating point representations (Gupta et al., 2015; Ott et al., 2018) or aggressive binarization (Courbariaux et al., 2017) can be alternatives. Compact models are usually faster to decode; studies on quantization (Lin et al., 2016) and average attention networks (Xiong et al., 2018) address this topic.

To the best of our knowledge, none of the above work has attempted to combine multi-model parameter tying, knowledge distillation, and dynamic layer selection for obtaining and exploiting highly-compressed and flexible deep neural models.

## 3   MULTI-LAYER SOFTMAXING

### 3.1   PROPOSED METHOD

Figure 1 gives a simple overview of the concept of multi-layer softmaxing, taking a generic model as an example. The rightmost 4-layer model takes an input and passes it through 4 layers[2] before a softmax layer to predict the output. Typically, one would apply softmax to the 4th layer only, compute loss, and then back-propagate gradients in order to update parameters. Instead, we propose to apply softmax to each layer, aggregate the computed losses, and then back-propagate. This enables us to choose any layer combination during decoding instead of only the topmost layer.

Extending this to a multi-layer encoder-decoder model is straightforward. In encoder-decoder models, the encoder comprises an embedding layer for the input (source language for NMT) and $N$ stacked transformation layers. The decoder consists of an embedding layer and a softmax layer for generating the output (target language for NMT) along with $M$ stacked transformation layers. Let $X$ be the input to the $N$-layer encoder, $Y$ the anticipated output of the $M$-layer decoder as well as the input to the decoder (for training), and $\hat{Y}$ the predicted output by the decoder. Algorithm 1 shows the pseudo-code for our proposed method. The line 3 represents the process done by the $i$-th encoder layer, $L_i^{enc}$, and the line 5 does the same for the $j$-th decoder layer, $L_j^{dec}$. In simple words, we compute a loss using the output of each of the $M$ decoder layers which in turn is computed using

---

[2]We make no assumptions about the nature of the layers.

the output of each of the $N$ encoder layers. In line 10, the $N \times M$ losses are aggregated[3] before back-propagation. Henceforth, we will refer to this as the *Tied-Multi model*.

For a comparison, the vanilla model is formulated as follows: $dec_j = L_j^{dec}(dec_{j-1}, enc_N)$, $\hat{Y} = softmax(dec_M)$, and $overall\_loss = cross\_entropy(\hat{Y}, Y)$.

## 3.2 EXPERIMENTAL SETUP

We evaluated the following two types of models on both translation quality and decoding speed.

**Vanilla models:** 36 vanilla models with 1 to 6 encoder and 1 to 6 decoder layers, each trained referring only to the last layer for computing loss.

**Tied-Multi model:** A single tied-multi model with $N = 6$ encoder and $M = 6$ decoder layers, trained by our multi-layer softmaxing.

We experimented with the WMT18 English-to-German (En→De) translation task. We used all the parallel corpora available for WMT18, except ParaCrawl corpus,[4] consisting of 5.58M sentence pairs as the training data and 2,998 sentences in newstest2018 as test data. The English and German sentences were pre-processed using the `tokenizer.perl` and `truecase.perl` scripts in `Moses`.[5] The true-case models for English and German were trained on 10M sentences randomly extracted from the monolingual data made available for the WMT18 translation task, using the `train-truecaser.perl` script available in `Moses`.

Our multi-layer softmaxing method was implemented on top of an open-source toolkit of the Transformer model (Vaswani et al., 2017) in the version 1.6 branch of `tensor2tensor`.[6] For training, we used the default model settings corresponding to `transformer_base_single_gpu` in the implementation, except what follows. We used a shared sub-word vocabulary of 32k determined using the internal sub-word segmenter of tensor2tensor and trained the models for 300k iterations. We trained the vanilla models on 1 GPU and our tied-multi model on 2 GPUs with batch size halved to ensure that both models see the same amount of training data. We averaged the last 10 checkpoints saved every after 1k updates, and decoded the test sentences, fixing a beam size[7] of 4 and length penalty of 0.6. We evaluated our models using the BLEU metric (Papineni et al., 2002) implemented in sacreBLEU (Post, 2018).[8] Prior to evaluation, the decoded test set was post-processed using the `detokenizer.perl` and `detruecase.perl` scripts in `Moses`. We also report the time (in seconds) consumed to translate the test set, which includes times for the model instantiation, loading the checkpoint, sub-word splitting and indexing, decoding, and sub-word de-indexing and merging, whereas times for detokenization are not taken into account.

Note that we did not use any development data for two reasons. First, we train all models for the same number of iterations.[9] Second, we use checkpoint averaging before decoding, which does not require a development set unlike early stopping. We adopted this approach, because it is known to give the best results for NMT using the Transformer implementation we use (Vaswani et al., 2017).

---

[3]We averaged multiple losses in our experiment, but there are a number of options, such as weighted averaging.

[4]`http://www.statmt.org/wmt18/translation-task.html`
We excluded ParaCrawl following the instruction on the WMT18 website: "BLEU score dropped by 1.0" for this task.

[5]`https://github.com/moses-smt/mosesdecoder`

[6]`https://github.com/tensorflow/tensor2tensor`

[7]One can realize faster decoding by narrowing down the beam width. This approach is orthogonal to ours and in this paper we do not insist which is superior to the other.

[8]`https://github.com/mjpost/sacreBLEU`
BLEU+case.mixed+lang.en-de+numrefs.1+smooth.exp +test.wmt18+tok.13a+version.1.3.7

[9]This enables a fair comparison, because it ensures that each model sees roughly the same number of training examples.

| | BLEU score | | | | | | | | | | | | Decoding time (sec) | | | | | |
| | 36 vanilla models | | | | | | Single tied-multi model | | | | | | | | | | | |
| $n\backslash m$ | 1 | 2 | 3 | 4 | 5 | 6 | 1 | 2 | 3 | 4 | 5 | 6 | 1 | 2 | 3 | 4 | 5 | 6 |
|---|---|---|---|---|---|---|---|---|---|---|---|---|---|---|---|---|---|---|
| 1 | 26.3 | 30.3 | 31.9 | 32.2 | 32.4 | 32.9 | 23.2 | 28.6 | 30.5 | 30.8 | 31.2 | 31.5 | 91.3 | 112.4 | 150.9 | 180.2 | 218.2 | 254.2 |
| 2 | 28.6 | 32.5 | 33.1 | 33.3 | 33.5 | 33.2 | 26.5 | 31.5 | 33.0 | 33.6 | 33.8 | 34.0 | 92.8 | 112.7 | 148.5 | 178.7 | 224.5 | 255.5 |
| 3 | 29.2 | 32.6 | 33.6 | 34.4 | 34.3 | 34.1 | 27.8 | 32.5 | 33.9 | 34.6 | 34.7 | 34.7 | 92.3 | 113.4 | 151.1 | 188.7 | 240.6 | 259.4 |
| 4 | 29.8 | 33.6 | 34.3 | 34.7 | 34.4 | 34.5 | 28.3 | 33.0 | 34.3 | 34.8 | 34.9 | 34.9 | 92.4 | 114.4 | 151.5 | 193.7 | 231.9 | 262.4 |
| 5 | 30.7 | 33.9 | 34.6 | 35.5 | 34.4 | 35.0 | 28.6 | 33.1 | 34.5 | 34.8 | 35.0 | 35.1 | 94.4 | 113.4 | 161.5 | 194.7 | 241.5 | 261.5 |
| 6 | 30.8 | 34.0 | 34.4 | **35.7** | 35.0 | 35.0 | 28.7 | 33.1 | 34.6 | 34.7 | 34.9 | 35.0 | 93.2 | 114.9 | 158.3 | 203.3 | 246.3 | 266.9 |

Table 1: BLEU scores of 36 separately trained vanilla models and our single tied-multi model used with $n$ ($1 \leq n \leq N$) encoder and $m$ ($1 \leq m \leq M$) decoder layers. One set of decoding times is also shown given the fact that a vanilla model with $n$ encoder and $m$ decoder layers and our tied-multi model have identical shapes when used with $n$ encoder and $m$ decoder layers for decoding.

## 3.3 RESULTS

Table 1 summarizes the BLEU scores and the decoding times of all the models, showing the cost-benefit property of our tied-multi model in comparison with the results of the corresponding 36 vanilla models.

Even though the objective function for the tied-multi model is substantially more complex than the one for the vanilla model, when performing decoding with the 6 encoder and 6 decoder layers, it achieved a BLEU score of 35.0, which is approaching to the best BLEU score of 35.7 given by the vanilla model with 6 encoder and 4 decoder layers.[10] Note that when using a single encoder layer and/or a single decoder layer, the vanilla models gave significantly higher BLEU score than the tied-multi model. However, when the number of layers is increased, there is no significant difference between the two types of models: less than 1.0 BLEU point differences.

Regarding the cost-benefit property of our tied-multi model, two points must be noted:

- BLEU score and decoding time increase only slightly, when we use more encoder layers.
- The bulk of the decoding time is consumed by the decoder, since it works in an auto-regressive manner. We can substantially cut down decoding time by using fewer decoder layers which does lead to sub-optimal translation quality.

One may argue that training a single vanilla model with optimal number of encoder and decoder layers is enough. However, as discussed in Section 1, it is impossible to know a priori which combination is the best. More importantly, a single vanilla model cannot suffice diverse cost-benefit demands and cannot guarantee the best translation for any input (see Section 4.1). Recall that we aim at a flexible model and that all the results in Table 1 have been obtained using a single tied-multi model, albeit using different number of encoder and decoder layers for decoding.

## 3.4 ANALYSIS AND DISCUSSION

We conducted an analysis from the perspective of training times and model sizes, in comparison with vanilla models.

**Training Time:** Given that all our models were trained for the same number of iterations, we compared the training times between vanilla and tied-multi models. As a reference, we use the vanilla model with 6 encoder and 6 decoder layers. The total training time for all the 36 vanilla models was 25.5 times[11] that of the reference model. In contrast, the training time for our tied-multi model was about 9.5 times that of the same reference model. Unexpectedly, training a tied-multi model was much more computationally efficient than independently training all the 36 vanilla models with different number of layers.

**Model Size:** The number of parameters of our tied-multi model is exactly the same as the vanilla model with $N$ encoder and $M$ decoder layers. If we train a set of vanilla models with different

---

[10]We consider this as an oracle, since we have no prior information about the best layer combination.

[11]We measured the collapsed time for a fair comparison, assuming that all vanilla models were trained on a single GPU one after another, even though one may be able to use multiple GPUs to train the 36 vanilla models in parallel.

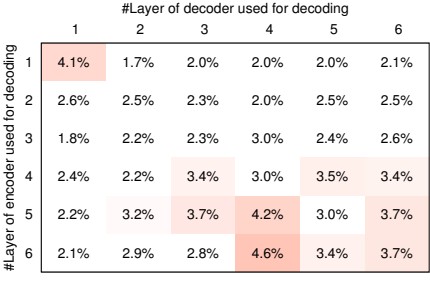
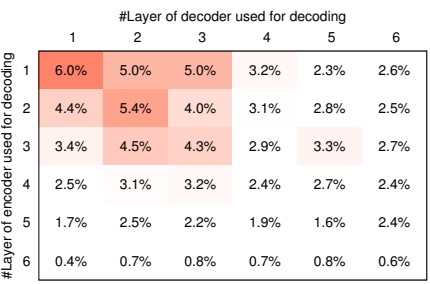

(a) 36 vanilla models.    (b) Single tied-multi model.

Figure 2: Distribution of oracle translations determined by chrF scores between reference and each of the hypotheses derived from the 36 combinations of encoder and decoder layers (2,998 sentences).

numbers of encoder and decoder layers, we end up with significantly more parameters. For instance, in case of $N = M = 6$ in our experiment, we have 25.2 times more parameters: a total of 4,607M for the 36 vanilla models against 183M for our tied-multi model. We discuss the possibility of further model compression in Section 5.

## 4 DYNAMIC LAYER SELECTION

To better understand the nature of our proposed method, we analyzed the distribution of oracle translations within 36 translations generated by each of the vanilla and our tied-multi models. Having confirmed that a single encoder-decoder layer combination cannot guarantee the best translation for any input, we tackled an advanced problem: designing a mechanism for dynamically selecting one layer combination prior to decoding.

### 4.1 DECODING BEHAVIOR OF TIED-MULTI MODEL

Let $(n, m)$ be an encoder-decoder layer combination of a given model with $n$ encoder and $m$ decoder layers. The oracle layer combination for an input sentence was determined by measuring the quality of the translation derived from each layer combination. We used a reference-based metric, chrF (Popović, 2016), since it has been particularity designed to sentence-level translation evaluation and was shown to have relatively high correlation with human judgment of translation quality for the language pair involved in our experiment (Ma et al., 2018). In cases where multiple combinations have the highest score, we chose the fastest one following the overall trend of decoding time (Table 1). Formally, we considered a combination $(n_1, n_2)$ is faster than another combination $(n_2, m_2)$ if the following holds.

$$(n_1, m_1) < (n_1, m_2) \equiv m_1 < m_2 \vee (m_1 = m_2 \wedge n_1 < n_2) \qquad (1)$$

Figure 2 presents the distribution of oracle layer combinations for the vanilla and our tied-multi models. A comparison between the two distributions revealed that the shallower layer combinations in our tied-multi model often generates better translations than deeper ones unlike the vanilla models, thanks to the multi-layer softmaxing method, despite the lower corpus-level BLEU scores. This sharp bias towards shallower layer combinations suggests the potential reduction of decoding time by dynamically selecting the layer combination per input sentence prior to decoding, ideally without performance drop.

In what follows, we present our method for dynamic selection of layer combination and experiments.

### 4.2 METHOD

We formalize the encoder-decoder layer combination selection with a supervised learning approach where the objective is to minimize the following loss function (2).

$$\arg\min_{\theta} \frac{1}{|X|} \sum_{x^i \in X} \mathcal{L}(f(x^i; \theta), y_k^i), \qquad (2)$$

where $x^i$ is the $i$-th input sentence ($1 \leq i \leq |X|$), $y_k^i$ is the translation for $x^i$ derived from the $k$-th layer combination ($1 \leq k \leq K = N \times M$) among all $K$ possible combinations, $f$ is the model with parameters $\theta$, and $\mathcal{L}$ is a loss function. Assuming that the independence of target labels (layer combinations) for a given input sentence allows for ties, the model is able to predict multiple layer combinations for the same input sentence.

The model $f$ with parameters $\theta$ implemented in our experiment is a multi-head self-attention neural network inspired by Vaswani et al. (2017). The number of layers and attention heads are optimized during a hyper-parameter search, while the feed-forward layer dimensionality is fixed to 2048. Input sequences of tokens are mapped to their corresponding embeddings, initialized by the embedding table of the tied-multi NMT model. A specific token is appended to each input sequence before being fed to the classifier. This token is finally fed during the forward pass to the output linear layer similarly to BERT (Devlin et al., 2019) for sentence classification. The output linear layer has $K$ dimensions, allowing to output as many logits as there are layer combinations in the tied-multi NMT model. Finally, a sigmoid function outputs probabilities for each layer combination among the $K$ possible combinations.

The parameters $\theta$ of the model are learned using mini-batch stochastic gradient descent with Nesterov momentum (Sutskever et al., 2013) and the loss function $\mathcal{L}$, implemented as a weighted binary cross-entropy (BCE) function detailed in (3) and averaged over the batch.

$$\mathcal{L}_{\text{BCE}_k} = -w_k \left[ \delta_k y_k \cdot \log \sigma(x) + (1 - y_k) \cdot \log(1 - \sigma(x)) \right], \tag{3}$$

where $\delta_k = (1 - p(y_k))^\alpha$ is the weight given to the $k$-th class based on class distribution prior knowledge and $\sigma$ is the sigmoid function. During our experiment, we have found that the classifier tends to favor recall in detriment to precision. To tackle this issue, we introduce another loss using an approximation of the macro $F_\beta$ implemented following (4).

$$\mathcal{L}_{F_\beta} = 1 - \left[ (1 + \beta^2) \cdot \frac{P \cdot R}{(\beta^2 \cdot P) + R} \right] \tag{4}$$

$$\text{with } P = \frac{\mu}{\sum_k \hat{y_k}}, R = \frac{\mu}{\sum_k y_k} \text{ and } \mu = \sum_k (\hat{y_k} \cdot y_k),$$

where $\hat{y_k}$ is the output of $\sigma(x)$ for the $k$-th dimension. The final loss function is the linear interpolation of $\mathcal{L}_{\text{BCE}}$ averaged over the $K$ classes and $\mathcal{L}_{F_\beta}$ with parameter $\lambda$: $\lambda \times \mathcal{L}_{\text{BCE}} + (1 - \lambda) \times \mathcal{L}_{F_\beta}$. We tune $\alpha$, $\beta$, and $\lambda$ during the classifier hyper-parameter search based on the validation loss.

## 4.3 EXPERIMENTS

The layer combination classifier was trained on a subset of the tied-multi NMT model training data presented in Section 3.2 containing 5.00M sentences, whereas the remaining sentences compose a validation and a test set containing approximately 200k sentences each. The two latter subsets were used for hyper-parameter search and evaluation of the classifier, respectively. To allow for comparison and reproducibility, the final evaluation of the proposed approach in terms of translation quality and decoding speed were conducted on the official WMT development (newstest2017, 3,004 sentences) and test (newstest2018, 2,998 sentences) sets; the latter is the one also used in Section 3.2.

The training, development, and test sets were translated with each layer combination of the tied-multi NMT model. Each source sentence was thus aligned with 36 translations whose quality were measured by the chrF metric. Because several combinations can lead to the best score, the obtained dataset was labeled with multiple classes (36 layer combinations) and multiple labels (ties with regard to the metric). During inference, the ties were broken by selecting the layer combination with the highest value given by the sigmoid function, or backing-off to the deepest layer combination (6, 6) if no output value reaches 0.5. This tie breaking method differs from the oracle layer selection presented in Equation (1) and in Figure 2 which prioritizes shallowest layer combinations. In this experiment, decoding time was measured by processing one sentence at a time instead of batch decoding, the former being slower compared to the latter, but leading to precise results. The decoding times were 954s and 2,773s when using a (1,1) and a (6,6) layer combination, respectively. By selecting the fastest encoder-decoder layer combinations according to an oracle, the decoding times went down to 1,918s and 1,812s for the individual and tied-multi models, respectively. However, our objective is to be faster than default setting, that is, where one would choose a (6,6) combination.

| Classifier | Fine-tuning | Speed (s) | BLEU |
|---|---|---|---|
| Baseline (tied (6,6)) | | 2,773 | 35.0 |
| Oracle (tied) | | 1,812 | 42.1 |
| 8 layers, 8 heads | x | 2,736 | 35.0 |
| 2 layers, 4 heads | x | 2,686 | 34.8 |
| 2 layers, 4 heads | | 2,645 | 34.7 |
| 4 layers, 2 heads | | 2,563 | 34.3 |

Table 2: Dynamic layer combination selection results in decoding speed (in seconds, batch size of 1) and BLEU, including the baseline and oracle for the WMT newstest2018 corpus using the tied-multi model architecture.

Several classifiers were trained before being evaluated on the WMT test set, with or without fine-tuning on the WMT development set. The results obtained in terms of corpus-level BLEU and decoding speed are presented in Table 2. Some classifiers maintain the translation quality (top rows), whereas others show quality degradation but further gain in decoding speed (bottom rows).

## 4.4 ANALYSIS

The classification results show that gains in decoding speed are possible with an a-priori decision for which encoder-decoder combination to select, based on the information contained in the source sentence only. However, no BLEU gains are observed in the results presented in Table 2 and a trade-off between decoding speed and translation quality is present. Our best configuration for decoding speed reaches a 210s decrease but leads to a 0.7 point BLEU degradation. In addition, only 37s are gained when preserving the translation quality compared to the baseline configuration. The layer combination oracle indicates substantial gains both in terms of BLEU (7 points) and decoding speed (961s). These oracle results motivate possible future work in layer combination prediction for the tied-multi NMT model.

## 5 FURTHER MODEL COMPRESSION

We examined the combination of our multi-layer softmaxing approach with another parameter-tying method in neural networks, called *recurrent stacking* (RS) (Dabre & Fujita, 2019), complemented by *sequence-level knowledge distillation* (Kim & Rush, 2016), a specific type of knowledge distillation (Hinton et al., 2015). We demonstrate that these existing techniques help reduce the number of parameters in our model even further.

## 5.1 DISTILLATION INTO A RECURRENTLY STACKED MODEL

In Section 2, we have discussed several model compression methods orthogonal to multi-layer softmaxing. Having already compressed $N \times M$ models with our approach, we consider further compressing it using RS. However, models that use RS layers tend to suffer from performance drops due to the large reduction in the number of parameters. As a way of compensating the performance drop, we apply sequence-level knowledge distillation.

First, we decode all source sentences in the training data to obtain a pseudo-parallel corpus containing distillation target sequences. By forward-translating the data, we create soft-targets for the child model which makes learning easier and hence is able to mimic the behavior of the parent model. Then, an RS child model is trained with multi-layer softmaxing on the generated pseudo-parallel corpus. Among a variety of distillation techniques, we chose the simplest one to show the impact that distillation can have in our setting, leaving an extensive exploration of more complex methods for the future.

## 5.2 EXPERIMENT

We conducted an experiment to show that RS and sequence distillation can lead to an extremely compressed tied-multi model which no longer suffers from performance drops. We compared the following four variations of our tied-multi model trained with multi-layer softmaxing.

| | $n\backslash m$ | Tied-multi model | | | | | | Tied-multi RS model | | | | | |
|---|---|---|---|---|---|---|---|---|---|---|---|---|---|
| | | 1 | 2 | 3 | 4 | 5 | 6 | 1 | 2 | 3 | 4 | 5 | 6 |
| | 1 | 23.2 | 28.6 | 30.5 | 30.8 | 31.2 | 31.5 | 25.7 | 29.8 | 30.6 | 30.8 | 30.7 | 30.9 |
| | 2 | 26.5 | 31.5 | 33.0 | 33.6 | 33.8 | 34.0 | 28.5 | 32.3 | 32.9 | 33.0 | 33.1 | 33.2 |
| without | 3 | 27.8 | 32.5 | 33.9 | 34.6 | 34.7 | 34.7 | 29.2 | 32.9 | 33.6 | 33.8 | 33.6 | 33.5 |
| distillation | 4 | 28.3 | 33.0 | 34.3 | 34.8 | 34.9 | 34.9 | 29.3 | 33.2 | 33.7 | 33.9 | 33.6 | 33.7 |
| | 5 | 28.6 | 33.1 | 34.5 | 34.8 | 35.0 | 35.1 | 29.4 | 33.2 | 33.7 | 33.9 | 33.9 | 34.0 |
| | 6 | 28.7 | 33.1 | 34.6 | 34.7 | 34.9 | 35.0 | 29.2 | 33.2 | 33.7 | 33.9 | 34.0 | 33.8 |
| | 1 | 30.1 | 34.0 | 35.1 | 35.3 | 35.6 | 35.7 | 31.2 | 33.5 | 34.1 | 34.2 | 34.3 | 34.3 |
| | 2 | 33.4 | 35.8 | 36.6 | 36.8 | 37.1 | 37.3 | 33.7 | 35.5 | 35.7 | 35.7 | 35.8 | 35.8 |
| with | 3 | 34.7 | 36.5 | 37.0 | 37.4 | 37.4 | 37.5 | 34.1 | 35.8 | 36.1 | 36.1 | 36.2 | 36.2 |
| distillation | 4 | 35.2 | 36.8 | 37.2 | 37.4 | 37.5 | 37.5 | 34.3 | 36.0 | 36.2 | 36.2 | 36.3 | 36.3 |
| | 5 | 35.5 | 36.9 | 37.1 | 37.4 | 37.5 | 37.6 | 34.5 | 36.1 | 36.2 | 36.3 | 36.3 | 36.3 |
| | 6 | 35.5 | 37.0 | 37.2 | 37.5 | 37.6 | 37.6 | 34.6 | 36.1 | 36.2 | 36.2 | 36.3 | 36.2 |

Table 3: BLEU scores of a total of four NMT models: the tied-multi model with (left block) and without (center and right blocks) RS layers, each trained with (top block) and without (bottom block) sequence distillation.

**Tied-multi model:** A model that does not share the parameters across layers, trained on the original parallel corpus.

**Distilled tied-multi model:** The same model as above but trained on the pseudo-parallel corpus.

**Tied-multi RS model:** A tied-multi model that uses RS layers, trained with the original parallel corpus.

**Distilled tied-multi RS model:** The same model as above but trained on the pseudo-parallel corpus.

Note that we defrayed much higher cost for training the distilled models than training models directly on the original parallel corpus. First, we trained 5 vanilla models with 6 encoder and 6 decoder layers, because the performance of distilled models is affected by the quality of parent models, and NMT models vary vastly in performance (around 2.0 BLEU) depending on parameter initialization. We then decode the entire training data (5.58M sentences) with the one[12] with the highest BLEU score on the newstest2017 (used in Section 4.3) in order to generate pseudo-parallel corpus for sequence distillation. Nevertheless, we consider that we can fairly compare the performance of the above four models, since they were trained only once with a random parameter initialization, without seeing the test set.

Table 3 gives the BLEU scores for all models. Comparing top-left and top-right blocks of the table, we can see that the BLEU scores for RS models are higher than their non-RS counterparts when using fewer than 3 decoder layers. This shows the benefit of RS layers despite the large parameter reduction. However, the reduction in parameters negatively affects (up to 1.3 BLEU points) when decoding with more decoder layers, confirming the limitation of RS as expected.

Comparing the scores of the top and bottom halves of the table, we can see that distillation dramatically boosts the performance of the shallower encoder and decoder layers. For instance, without distillation, the tied-multi model gave a BLEU of 23.2 when decoding with 1 encoder and 1 decoder layers, the same layer combination reaches 30.1 BLEU through distillation. Given that RS further improves performance using lower layers, the BLEU score increases to 31.2. As such, distillation enables decoding using fewer layers without substantial drops in performance. Furthermore, the BLEU scores did not vary significantly when the layers deeper than 3 were used, meaning that we might as well train shallower models using distillation. The performance of our final model, i.e., the distilled tied-multi RS model (bottom-right), was significantly lower (up to 1.5 BLEU points) similarly to its non-distilled version. However, given that it outperforms our original tied-multi model (top-left) in all the encoder-decoder layer combinations, we conclude that we can obtain a substantially compressed model with better performance.

---

[12]Ensemble of multiple models is commonly used for distillation, but we used a single model to save decoding time.

| Model(s) | Parameters | Relative size |
|---|---|---|
| 36 vanilla models | 4,608M | 25.16 |
| Single tied-multi model | 183M | 1.00 |
| 36 RS models | 2,623M | 14.33 |
| Single tied-multi RS model | 73M | 0.40 |

Table 4: Model sizes for different encoder-decoder layer combinations. The relative size is calculated regarding the tied-multi model as a standard. Similarly to "36 vanilla models," "36 RS models" represents the total number of parameters of all RS models.

## 5.3 ANALYSIS

We now analyze model size and decoding speed resulted by applying RS and knowledge distillation. Note that RS has no effect on training time because the computational complexity is the same with or without it.

**Model Size:** Table 4 gives the sizes of various models that we have trained and their ratio with respect to the tied-multi model. Training vanilla and RS models with 36 different encoder-decoder layer combinations required 25.2 and 14.3 times the number of parameters of a single tied-multi model, respectively. Although RS led to some parameter reduction, combining RS with our tied-multi model resulted a further compressed single model: 0.40 times that of the single tied-multi model without RS. This model has 63.2 times and 36.0 times fewer parameters than all the individual vanilla and RS models, respectively. Given that knowledge distillation can reduce the performance drops due to RS (see Table 3), we believe that combining it with this approach is an effective way to compress a large number of models into one.

**Decoding Speed:** Although we do not give scores due to lack of space, we observed that greedy decoding is faster than beam decoding but suffers from significantly reduced scores of around 2.0 BLEU. By using our distilled models, however, greedy decoding reduced the scores only by 0.5 BLEU compared to beam decoding. This happens because we have used translations generated by beam decoding as target sentences for knowledge distillation, which has the ability to loosely distill beam search behavior into greedy decoding behavior. For instance, greedy decoding with the distilled tied-multi RS model with 2 encoder and 2 decoder layers resulted in a BLEU score of 35.0 in 66.1s. In comparison, beam decoding with the tied-multi model without RS and distillation with 5 encoder and 6 decoder layers led a BLEU score of 35.1 in 261.5s (Table 1), showing that comparable translation quality is obtained with a factor of 4 in decoding time when using RS and distillation. Even though we intended to minimize performance drops besides the model compression, we obtained an unexpected benefit in terms of faster decoding through greedy search without a significant loss in translation quality.

## 6 CONCLUSION

In this paper, we have proposed a novel procedure for training encoder-decoder models, where the softmax function is applied to the output of each of the $M$ decoder layers derived using the output of each of the $N$ encoder layers. This compresses $N \times M$ models into a single model that can be used for decoding with a variable number of encoder ($1 \leq n \leq N$) and decoder ($1 \leq m \leq M$) layers. This model can be used in different latency scenarios and hence is highly versatile. We have experimented with NMT as a case study of encoder-decoder models and given a cost-benefit analysis of our method. We have proposed and evaluated two orthogonal extensions and show that we can (a) dynamically choose layer combinations for slightly faster decoding and (b) further compress models using recurrent stacking with knowledge distillation.

For further speed up in decoding as well as model compression, we plan to combine our approach with other techniques, such as those mentioned in Section 2. Although we have only tested our idea for NMT, it should be applicable to other tasks based on deep neural networks.

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
