# OpenReview forum: "Balancing Cost and Benefit with Tied-Multi Transformers"
_ICLR.cc/2020/Conference — Reject_

### Official Review · AnonReviewer1 · 2019-10-21
**Official Blind Review #1**

**Rating:** 1

**Review:**

** Summary **
In this paper, the authors propose a new variant of Transformer called Tied-multi Transformer. Given such a model with an N-layer encoder and an M-layer decoder, it is trained with M*N loss functions, where each combination of the nth-layer of the encoder and the mth-layer of the decoder is used to train an NMT model. The authors propose a way to dynamically select which layers to be used when a specific sentence comes.  At last, the authors also try recurrent stack and knowledge to further compress the models.

** Details **
1.	The first question is “why this work”:
a.	In terms of performance improvement, in Table 2, we can see that dynamic layer selection does not bring any improvement compared to baseline (Tied(6,6)). When compared Tied(6,6) to standard Transformer, as shown in Table 1, there is no improvement. Both are 35.0.
b.	In terms of inference speed, in Table 2, the method can achieve at most (2773-2563)/2998 = 0.07s improvement per sentence, which is very limited.
c.	In terms of training speed, compared to standard Transformer, the proposed method takes 9.5 time of the standard Transformer (see section 3.4, training time).
Therefore, I think that compared to standard Transformer, there is not a significant difference.
2.	 The authors only work on a single dataset, which is not convincing.
3.	In Section 5, what is the baseline of standard RS + knowledge distillation?


**Experience Assessment:**

I have published in this field for several years.

**Review Assessment: Checking Correctness Of Derivations And Theory:**

I carefully checked the derivations and theory.

**Review Assessment: Checking Correctness Of Experiments:**

I carefully checked the experiments.

**Review Assessment: Thoroughness In Paper Reading:**

I read the paper thoroughly.

---

> ### Author Response · Authors · 2019-11-12
> **Efficiency is why our work is important. Our work can be a starting point.**
>
> We thank you for your review and for taking the time to read our paper thoroughly.
>
> Our responses to your questions are as follows:
>
> 1. You are right that a decoding speed gain of 0.07s per sentence is limited, but this gain is obtained with a system as performant as the multi-tied model without routing at the corpus level (BLEU 35.0). With non-significant loss in BLEU, i.e. 34.7 for instance, a larger decoding speed gain is measured. Additionally, for a large pool of data to decode, the gain in decoding time is worthwhile. The main purpose of the dynamic layer selection approach is to draw attention to the concept of flexible decoding. One of the results is that, using source sentences only as information given to the classifier, the task is difficult. Regarding the training time of the tied-multi model, flexibility and hence faster decoding is not possible with a standard transformer. We will have to train several transformer models with different layer configurations to achieve the level of flexibility reached by the tied-multi model. While training 36 models takes needs  25.5 times the training time of a 6-6 model, training our flexible model takes only 9.5 times the training time of a 6-6 model. As such our work does have merit.
>
> 2. We understand that working on multiple datasets would make our work more convincing but we do not expect our results to be dataset dependent as we make no assumption about the type of data used. Results on other datasets will be included in future manuscripts.
>
> 3. We trained the RS+KD model but did not include it in the paper because we considered it fair to compare multilayer softmaxed models only. Nevertheless the BLEU score of RS+KD is 36.1 which is not statistically significantly different from the best BLEU of 36.3 of the multilayer model.

---

### Official Review · AnonReviewer2 · 2019-10-23
**Official Blind Review #2**

**Rating:** 1

**Review:**

This work proposes a way to reduce the latencies incurred in inference for neural machine translation. Basic idea is to train a model with softmax attached to each output of decoder layers, and computes a loss by aggregating the cross entropy losses over the softmaxes. During inference, it could either use one of the softmax or train an additional model which dynamically selects softmaxes given an input string. Experimental results show that it is possible to reduce latencies by trading off the translation qualities measured by BLEU. Dynamically selection did not show any gains in latencies, though, this work empirically shows potential gains in oracle studies. This work further shows that the model could be compressed further by knowledge distillation.

I have several concerns to this work and I'd recommend rejecting this submission.

- One of the problems of this paper is presentation. This work basically combines three work together as a single paper, i.e., section 3 for the basic model, section 4 for dynamic selection and section 5 for distillation, with each section describing a separate experiment. I'd strongly suggest the author to focus on the main point, e.g., dynamic selection, and present the basic model and dynamic selection. Experiments should be presented in a single section for brevity.

- Similarly, this work should have been submitted when meaningful gains were observed in the dynamic selection method, given that the proposal is somewhat new. Otherwise, I don't find any merits to see this accepted in ICLR, given the rather negative results in section 4.

- The description in section 4.2 is totally messed up. x^i and y^i_k are strings since they are an input sentence and an output sentence, respectively,. However, they are treated as scalars in Equation 3 by multiplied with \delta_k, subtracted from 1 and taking sigmoid through \sigma. I strongly suggest authors to carefully check variables used in the equations and the description in the section.

- The authors claim that the use of knowledge distillation is novel. However, it is already widely known in the research community and I don't think it is worthy to keep it as a single section. It could have been described as a yet another experiment in a single experimental section.

Other comment:

- Although this paper claims that attaching a softmax for each output layer is new, there is a similar work in language modeling, though the motivation is totally different.

  Direct Output Connection for a High-Rank Language Model, Sho Takase, Jun Suzuki and Masaaki Nagata, EMNLP 2019.

- In section 3.4, this paper claims that the training of all 36 models took 25.5 more time, but took 9.5 more time for a tied-model when compared with a basic 6-layer Transformer. It is not clear to me whether this comparison is meaningful given that it might be possible to employ multiple machines to train 36 models.

**Experience Assessment:**

I have published in this field for several years.

**Review Assessment: Checking Correctness Of Derivations And Theory:**

I carefully checked the derivations and theory.

**Review Assessment: Checking Correctness Of Experiments:**

I assessed the sensibility of the experiments.

**Review Assessment: Thoroughness In Paper Reading:**

I read the paper at least twice and used my best judgement in assessing the paper.

---

> ### Author Response · Authors · 2019-11-12
> **Thank you. The reorganization will be done.**
>
> We thank you for your review and for taking the time to read our paper thoroughly.
>
> Our responses to your questions are as follows:
>
> 1. Thank you for your suggestion regarding the reorganization of the paper. The basic model, which includes the multi-softmax functions, is the main point of the paper. Dynamic layer selection and distillation are two extensions of the basic model which allow to improve decoding speed and further parameters reduction.
>
> 2. Although we did not report it, we found that it is actually quite hard to obtain gains as well as speed. We tried a large number of what we believed to be promising approaches and most of them failed to give gains. What we reported was the most promising one. Our hypothesis is that there is too much randomness in the behavior of our NMT models and while there are few gains, it should be worthwhile to take as much as we can get. Given that our routing method is simple we decided that it should be reported.
>
> 3. Sorry for totally messing up section 4.2. Indeed the variable names are wrong in Eq. 2.
>
> 4. The use of distillation in the context of flexibly decodable models that also use recurrently stacked layers is new but we do understand why it should be incorporated into another section.
>
> 5. We thank you for pointing us to the other related work but the motivation is quite different as you say and our claim regarding novelty is relevant only to NMT.
>
> 6. Regarding the training time we are comparing the total GPU computation hours.

---

### Official Review · AnonReviewer3 · 2019-10-23
**Official Blind Review #3**

**Rating:** 6

**Review:**


This paper proposes a novel procedure for training multiple Transformers with tied parameters which compresses multiple models into one enabling the dynamic choice of the number of encoder and decoder layers during decoding. The idea is simple and reasonable and the results are promising.

I have several questions about the paper:
	1. "This enables a fair comparison, because it ensures that each model sees roughly the same number of training examples." This is not a fair comparison. Note that those models are of very different size, and thus they may need different numbers of samples for training. For example, a 1-1 model should need much less data for training than a 6-6 model. If the number of training samples is ok for the 1-1 model, it might be insufficient for the 6-6 model. Therefore, I think development set is necessary for a fair comparison.

	2. I don't understand Eq. (3). What do x and y_k mean in this equation? Are they corresponding to x^I and y^i_k in Eq (2)? However, y^i_k in Eq. (2) is a translation, i.e., a text sentence, while y_k in Eq. (3) looks like a number in [0,1].


**Experience Assessment:**

I have published in this field for several years.

**Review Assessment: Checking Correctness Of Derivations And Theory:**

I carefully checked the derivations and theory.

**Review Assessment: Checking Correctness Of Experiments:**

I carefully checked the experiments.

**Review Assessment: Thoroughness In Paper Reading:**

I read the paper thoroughly.

---

> ### Author Response · Authors · 2019-11-12
> **Thank you**
>
> We thank you for your review and for taking the time to read our paper thoroughly. We are happy to read that you think that our work is promising.
>
> Our responses to your questions are as follows:
>
> 1. In the case of the implementation we used, our 6-6 model was trained for 300k iterations and we do agree that a 1-1 model should need fewer than 300k iterations. However, we observed that training a 1-1 model for much longer does not run into problems such as overfitting. The BLEU scores do not vary statistically significantly. We do agree that it might not give us the best 1-1 model and thereby limit fairness.
>
> 2. Sorry for the confusion. In Eq. 3, x refers to the logits prior to the sigmoid layer produced by the neural network and y_k is the reference class for a given input sentence.

---

### Decision · Program_Chairs · 2019-12-19

**Decision:**

Reject

**Comment:**

The paper proposed a method for training multiple transformers with tied parameters and enabling dynamic choice of the number of encoder and decoder layers. The method is evaluated in neural machine translation and shown to reduce decoding costs without compromising translation quality. The reviewers generally agreed that the proposed method is interesting, but raised issues regarding the significance of the claimed benefits and the quality of overall presentation of the paper. Based on a consensus reached in a post rebuttal discussion with the reviewers, I am recommending rejecting this paper.